# Tracing Emotional Evolution along Named Entity Topic

# **2 Chains: A Mechanistic Study of Chinese Social Media in the**

# 3 2025 Myanmar Earthquake

Changqi Dong<sup>1</sup>, Kaihang Zhang<sup>1</sup>, Jida Liu\*

School of Management, Harbin Institute of Technology, Harbin, 150001, China

Correspondence: Jida Liu (kittadada@yeah.net )

These authors contributed equally to this work.

ABSTRACT

7

10

18

This study examines how emotional responses to transboundary disasters are structured and propagated within digital discourse, using the 2025 Myanmar earthquake as a case. Drawing on a dataset of 139,473 Chinese Weibo posts collected from March 28 to April 25, we develop an emotion-entity coupling framework that integrates large language model-based emotion annotation with named entity recognition (NER) to construct a semantic-affective network. Rather than treating sentiment as a standalone attribute, this approach models emotion as a dynamic and relational process that flows through named entities, which serve as semantic anchors and emotional conduits. The analysis reveals distinct patterns in both temporal emotion dynamics and structural emotion transmission. While emotions such as fear and surprise dominated the discourse, positive sentiments, particularly those associated with humanitarian actors, formed localized zones of empathic resonance. The coupled emotion - entity network exposed asymmetric affective pathways, with certain entities acting as hubs of amplification, bridge nodes, or buffers in the transmission of emotional meaning. Subgraph analysis further highlighted how institutional memory, geographical proximity, and media narratives shaped the stability and flow of public sentiment. By reconceptualizing emotion as structurally embedded and semantically routed, this study offers both theoretical and methodological innovations in disaster risk communication. The proposed framework advances understanding of how empathy and public engagement are generated, distributed, and sustained in the digital age, particularly in response to disasters that cross national borders.

3132 Key words:

- Emotion-Entity Coupling
- Named Entity Recognition (NER)
- Social Media Analytics
- Emotion Diffusion
- Transboundary Disasters

1

### 1. Introduction

39

The global proliferation of social media platforms has transformed the landscape of disaster 41 communication, enabling real-time expression, diffusion, and amplification of public sentiment in the wake of crisis events (Alexander, 2014; Li et al., 2020; Erokhin and Komendantova, 2024). In 42 43 transboundary disasters, where the physical impacts transcend national borders or where distant 44 publics engage in empathetic reactions to catastrophes abroad, social media operates not only as 45 an information infrastructure but also as an affective interface (Tim et al., 2017; Kaufhold and 46 Reuter 2016; Samatan et al., 2020). Recent scholarship has increasingly recognized that emotional 47 responses shared online play a central role in shaping collective understanding, influencing 48 behavioral intentions, and constructing emergent forms of digital solidarity (Lüders et al., 2022; 49 Castellanos et al., 2025; Storey and O'Leary, 2024; Yin et al., 2024). Yet, the mechanisms by 50 which emotions propagate, transform, and embed themselves within evolving topic structures on 51 social media remain underexplored, particularly in contexts where the affected population and the 52 reacting public are geographically decoupled. 53 The 2025 Myanmar earthquake, one of the deadliest seismic events since the 2023 54 Turkey-Syria catastrophe and widespread regional impacts across Southeast Asia and southern China, exemplifies such a transnational event. Although the epicenter was located in Myanmar's 55 56 Sagaing Region, the tremors were felt across national boundaries, including in China's Yunnan 57 Province, where infrastructural damages and injuries were reported. Beyond the physical reach of 58 seismic waves, the disaster triggered waves of emotional resonance on Chinese social media, 59 especially Weibo, where users collectively expressed grief, concern, and solidarity. The 60 temporality and structure of these expressions did not emerge arbitrarily; rather, they were interwoven with specific topics, key entities (such as locations, institutions, or individuals), and 61 62 recurring themes. Understanding how emotions evolve within these semantic chains, and how they 63 are sustained, intensified, or redirected through interaction with named entities and emerging 64 discourse patterns. is essential for uncovering the latent cognitive infrastructures of digital disaster 65 response (Ma and Zheng, 2025; Stella, 2022). Traditional sentiment analysis approaches, which classify messages based on valence 66 67 (positive, negative, or neutral), offer limited insight into the nuanced emotional dynamics present

68 in post-disaster discourse (Bai and Yu, 2016; Shapouri et al., 2025; Ma et al., 2024). Emotions in 69 crisis communication are rarely monolithic. Instead, they shift across phases, from shock and 70 sorrow in the immediate aftermath to calls for justice, mobilization of aid, or expressions of 71 resilience in later stages (Han and Wang, 2022; Börner, 2020). While prior work has investigated 72 emotional trajectories over time (Ma et al., 2020; Chen et al., 2022; Xu et al., 2020; Zhang et al., 73 2024; Wang et al., 2024), few studies have linked emotional evolution to the semantic structure of 74 social media discourse, that is, the dynamic alignment between emotional expression and the 75 underlying topic chains composed of named entities, actions, and contextual cues. 76 To address this gap, this study adopts an integrated methodological framework that combines large language model-based fine-grained emotion annotation, and named entity recognition (NER). 77 Leveraging a dataset of 139,473 Weibo posts collected between March 28 and April 25, 2025, we 78 79 first examine the descriptive landscape of public discourse and emotional trends following the 80 Myanmar earthquake. We then identify topic clusters using a BERT model enhanced with a large 81 language model, and annotate each post with discrete emotional categories (e.g., sadness, 82 sympathy, anger, hope) (George and Sumathy, 2023; Zhang et al., 2025a). We then employ named entity recognition (NER) to extract key semantic components from each post, including references 83 84 to places, organizations, individuals, and disaster-related terms (Eligüzel et al., 2022; Qiu et al., 85 2023; Singh and Singh, 2023). These entities are used to construct a topic-focused linkage network, 86 in which nodes represent entities and edges represent their co-occurrence within the same textual 87 context (Misuracaet al., 2020; Khurana al., 2023). This network constitutes a semantic backbone 88 over which emotional signals can be mapped and traced. On this entity-based topic network, we 89 model emotional dynamics by assigning each node a dominant emotional attribute and recording 90 the temporal onset of that emotion. This allows us to visualize and analyze how emotions 91 propagate along semantic chains, whether they intensify, shift, or dissipate as discourse unfolds 92 over time. 93 This paper makes three key contributions. First, it advances a novel integration of 94 LLM-based emotion classification with entity-centered semantic modeling, enabling fine-grained 95 analysis of emotional trajectories along topic networks. Second, it develops a pathway-based view 96 of emotional evolution, identifying structural features of emotional diffusion such as 97 intensification points, transition paths, and resonance sub-networks. Third, it provides empirical

evidence of how digital publics in China engaged affectively with a foreign disaster, revealing patterns of transnational solidarity, symbolic alignment, and cognitive resonance.

#### 2. Literature review

Over the past decade, research on social media and disasters has evolved from descriptive accounts of platform usage toward more nuanced investigations into the affective dimensions of crisis communication (Jin et al., 2011; Mirbabaie et al., 2020). Emotions expressed online during disaster events are now recognized as more than psychological reactions; they constitute discursive resources that shape collective interpretations, mobilize responses, and reconfigure public spaces (Reuter and Kaufhold, 2018; Stieglitz et al., 2018). Within this trajectory, the study of emotional dynamics, how emotions emerge, intensify, and shift across time, has become an area of growing interest.

A substantial body of work has examined the temporal evolution of public emotions in disaster settings. Typically, these studies deploy sentiment analysis tools to track aggregate changes in affective valence (e.g., positive, negative, neutral) across different phases of a crisis. For instance, Lee et al. analyzed Twitter, YouTube, and Facebook data during the Sewol ferry disaster using machine learning methods, finding that public negative sentiment was closely linked to the government's disaster response policies (Lee et al., 2020). Similarly, Zheng et al. tracked the evolution of public sentiment in Wuhan, China within the first 12 weeks after the discovery of COVID-19 on Sina Weibo, a Chinese microblogging platform. They found that from confusion/fear to disappointment/depression, then to depression/anxiety, and finally to happiness/gratitude, this progression indexed the constantly changing emotional energy of digital medical citizens (Zheng et al., 2021). These approaches, while valuable, tend to treat emotional expression as temporally indexed but disconnected from semantic context, that is, emotions are tracked by time, but not by topic or referential structure. These approaches, while valuable, tend to treat emotional expression as temporally indexed but disconnected from semantic context, that is, emotions are tracked by time, but not by topic or referential structure.

To overcome these limitations, some scholars have introduced topic-based emotion analysis, attempting to link affective expressions to particular themes or concerns. Methods such as topic modeling (e.g., LDA, BERTopic) have been used to group semantically related posts, enabling

128129

137138

researchers to examine how different emotional tones map onto distinct thematic clusters (Zhang et al, 2025b). However, most of these studies operate at a high level of abstraction, using bag-of-words representations that neglect named entities, syntactic dependencies, or discourse structures. As a result, they often fail to capture how emotions are tied to specific actors, places, or events within the communicative landscape. By contrast, Named Entity Recognition (NER) has been widely adopted in disaster informatics for the extraction of structured information, such as locations affected, organizations involved, and individuals referenced in user-generated content (Sun et al., 2022; Li and Zhang, 2023; Hu et al., 2023). NER enhances the granularity of information retrieval and facilitates applications such as situational awareness dashboards, geospatial mapping of damage reports, and automated alert systems (Wilkh et al., 2025). Yet despite its widespread use, NER is rarely integrated with emotion analysis beyond simple co-occurrence statistics. There remains a notable gap in understanding how named entities function as semantic anchors or transmission points for emotional content, and whether certain types of entities (e.g., symbols of authority, vulnerable groups, frontline actors) are systematically associated with shifts in emotional states. Some recent work has begun to explore emotion propagation within social networks, particularly through the lens of emotional contagion. Studies in this area model how affective states spread across users via retweet patterns, follower graphs, or reply trees (Lwowski et al., 2018; Venkatesan et al., 2021; Murdock et al., 2024; Chu et al., 2024). These models often draw on epidemiological or diffusion frameworks, treating emotion as a transmissible unit that flows through interpersonal ties. While this line of research offers important insights into inter-user dynamics, it tends to overlook the semantic infrastructure of emotion transmission, that is, how the content of messages, especially the entities and themes they invoke, shapes the form and direction of emotional diffusion. Furthermore, although the concept of empathic communication has gained traction in disaster studies, it is frequently operationalized through lexical sentiment or aggregate trends, with limited attention to the structural features of discourse that enable empathy to emerge and stabilize. In the broader field of discourse and affect studies, increasing attention has been paid to how referential structures and symbolic anchors influence the emotional dynamics of public communication. Rather than treating emotions as isolated reactions, recent research has

160

165

168

173

183

highlighted how they are linked to, and often stabilized by, specific narrative forms, semantic roles, and symbolic markers (Gay, 2025). In the context of disaster communication, named entities such as "government agencies," or "international aid groups" often function as discursive nodes that structure how publics attribute responsibility, mobilize empathy, or express collective frustration. However, existing research predominantly employs text co-occurrence methods to extract key entities, a approach that tends to overlook the inherent relationships between entities within the text (Dai et al., 2024). This paper builds on such perspectives by treating entity chains not just as co-occurrence patterns but as emergent semantic pathways that channel and transform emotional currents in post-disaster discourse. This study seeks to extend this literature in two critical ways. First, it proposes a framework

that integrates LLM-based emotion annotation with entity-level topic chaining, allowing for the modeling of emotional evolution along named entity networks. Second, it treats emotion not simply as a label affixed to a post, but as a dynamic state that is shaped by and shapes the semantic structure of discourse. By tracing how emotions propagate, transform, and cluster within entity-based linkage graphs, the study provides a mechanism-oriented account of empathic communication during transboundary disasters. Such an approach not only enriches current understandings of emotion in crisis contexts but also offers methodological innovations relevant to the broader field of disaster risk reduction.

## 3. Data and Methodology

### 3.1 Data Collection and Preprocessing

To investigate the emotional and semantic dynamics of Chinese public response to the 2025 Myanmar earthquake, we constructed a large-scale social media dataset by collecting original microblog posts from Sina Weibo, China 's most prominent social media platform. Data acquisition was conducted using a combination of keyword filtering, platform-specific API access, and automated scripts. The keyword set included "缅甸地震 (Myanmar Earthquake)," "实皆省 (Sagaing Region),""中国救援 (Chinese rescue),""红十字会 (Red Cross)," and other related terms referencing the event, locations, and organizations involved. The data collection window was aligned with the active discussion period following the disaster, spanning from March 28 to April 25, 2025, as illustrated in Figure 1. March 28 marked

the occurrence of the earthquake, and our collection extended nearly one month to capture both the acute reaction phase and subsequent developments in public discourse. In total, 139,473 original microblog entries were retrieved and included for analysis after preprocessing.

Fig. 1 Data collection timeline for Weibo posts related to the 2025 Myanmar Earthquake.

Each post record contained both textual content and structured metadata fields, allowing for a multi-dimensional analysis. The metadata included: (1) User-related features: gender, follower count, number of followees, verification type, user category (e.g., media, individual, government), and location; (2) Post engagement indicators: number of reposts, comments, likes, and an aggregated engagement score (interaction count); (3) Content features: character count, posting time, and declared publishing location.

Data preprocessing followed a standardized pipeline (George and Baskar, 2024). Posts containing advertisements, duplicated content, or irrelevant hashtags were filtered out through rule-based and semantic screening. Text normalization was applied to remove redundant characters, emojis, and formatting symbols while preserving essential linguistic and emotional cues. Simplified Chinese character standardization was enforced to ensure consistency across texts.

The dataset was subsequently anonymized and encoded in UTF-8 format for model compatibility. All user identifiers were removed to preserve privacy in compliance with ethical guidelines. The cleaned and structured corpus formed the basis for subsequent analysis in both supervised model training and unsupervised pattern detection.

This structured dataset provided a robust foundation for modeling both content-level and actor-level dynamics in the disaster response discourse. With temporal resolution, entity-rich context, and interaction features embedded, the corpus allowed for the exploration of not only what was said, but also who said it, how widely it was diffused, and how sentiment evolved over time. Importantly, the inclusion of location metadata and declared publishing addresses made it

possible to examine the spatial diffusion of affective and thematic attention, offering a window into the translocal empathy mechanisms emerging from within China in response to an external crisis.

In addition to general descriptive statistics on user and content attributes (see Section 4.1), this dataset served as the input for fine-grained annotation tasks and subsequent large-scale emotion classification and named entity recognition, detailed in Sections 3.2 and 3.3.

### 3.2 Experimental Design

To model the interplay between semantic structure and emotional dynamics in disaster-related discourse, we designed a multi-stage experimental workflow that integrates human-machine hybrid annotation, dual-task model fine-tuning, and full-sample automated inference. The complete experimental pipeline is illustrated in Figure 2.

Fig. 2 Data collection timeline for Weibo posts related to the 2025 Myanmar Earthquake.

To begin, we randomly sampled 10% of the full dataset, approximately 13,947 Weibo posts, from the corpus of 139,473 entries. This sample was annotated using a hybrid strategy that combines the output of GLM-4-Flash, a large-scale Chinese generative language model, with manual refinement by trained annotators. For emotion classification, we adopted the seven-category framework from the Dalian University of Technology Chinese Emotional Ontology (DUTIR), which includes: Surprise, Sadness, Fear, Happiness, Anger, Disgust, and Good (Liu et al., 2023). Each post was assigned a single dominant emotional category based on overall tone and contextual meaning. In parallel, we manually labeled named entities using the

from tokenization to network construction.

standard BIO tagging format, including locations, organizations, government bodies, 234 disaster-related terms, and individuals. Based on this annotated dataset, we fine-tuned two BERT-based models using the 235 236 bert-base-chinese pretrained language model as backbone. The first model is a multi-class emotion 237 classifier trained using a softmax activation layer, while the second is a sequence labeling model 238 for named entity recognition (NER), using a Conditional Random Field (CRF) decoding layer. 239 These models were trained independently but applied jointly to the full dataset for automatic 240 labeling of all 139,473 posts. As a result, each post was assigned both an emotion label and a 241 corresponding set of extracted named entities. 242 We then constructed an emotion-entity coupled network by integrating the NER results with 243 the emotion classification output. In this network, each node represents a named entity extracted 244 from the corpus, and undirected edges denote entity co-occurrence within the same post. To 245 incorporate affective information, we aggregated emotion labels across all posts in which a given 246 entity appeared, and assigned each node a dominant emotion determined by frequency. When 247 multiple entities co-occurred in the same post, their emotional associations were also used to trace potential emotion transitions across the entity network. If entity A frequently appeared in 248 249 Sadness-labeled posts and co-occurred with entity B, which was often associated with Anger, a 250 directed emotion transition from A to B was inferred. By accumulating these transition patterns 251 across the corpus, we generated an emotion-informed entity graph capable of capturing the 252 structural pathways through which emotional content spreads within semantic discourse. 253 This experimental setup enables a granular, mechanism-based understanding of emotion 254 propagation in public responses to disasters. Rather than treating emotions as isolated labels, our 255 design embeds affective information within a referential and co-occurrence structure, revealing 256 how emotional meaning is entangled with semantic anchors in the digital communication of risk. 3.3 NER Principle in Myanmar Earthquake Study 257 258 To extract semantically rich references from disaster-related discourse, we implemented a 259 fine-tuned Named Entity Recognition (NER) model grounded in Chinese language structure and crisis-specific vocabulary. As illustrated in Figure 3, our approach follows a multi-step sequence 260

Fig. 3 Multi-step sequence of NER in Myanmar Earthquake Study.

Tokenized inputs are then fed into a BERT-CRF architecture fine-tuned on disaster-related corpora. The model assigns BIO-format tags, denoting entity boundaries as *Beginning, Inside*, or *Outside*, to each token. For example, in the sentence "缅甸 7.7 级地震造成重大伤亡", the tokens "缅甸" and "7.7 级地震" are labeled as B-LOC and B-EVENT respectively, while subsequent tokens such

The NER pipeline begins with text preprocessing and token segmentation of each Weibo post.

as "伤亡" are tagged as B-IMP, indicating disaster impact.

We defined seven named entity categories to reflect the complex referential landscape of disaster discourse: (1) Geography: places and regions (e.g., Mandalay, Sagaing); (2) Person: individuals or groups (e.g., victims, officials); (3) Organization: institutions involved (e.g., Red Cross, military); (4) Event/Outcome: natural events and consequences (e.g., earthquake, collapse, casualties); (5) Time: temporal markers (e.g., March 28, 72 hours); (6) Quantity/Value: numbers and measurements (e.g., magnitude 7.7, 300+ houses); (7) Other/Object: material items or abstract references (e.g., tents, medical supplies).

Following entity extraction, we constructed an entity co-occurrence network in which each node represents a named entity and edges connect entities appearing in the same post. Edge weights are determined by co-occurrence frequency, enabling a structural representation of how

disaster-related concepts, actors, and outcomes are discussed in proximity. This network serves as the semantic scaffold for subsequent emotion projection: each node is assigned a dominant emotional label based on the distribution of emotions in the posts where the entity appears, and directional emotion transitions between nodes are inferred from their co-mention patterns.

By structuring unstructured microblog discourse into an interpretable semantic graph, the NER process enables mechanism-based modeling of how public attention and meaning are organized in the wake of transboundary disaster events. This also supports the construction of topic-focused emotion pathways as elaborated in Section 3.2.

### 4. Results

#### 4.1 Temporal Dynamics and Post Characteristics

The temporal distribution of posts related to the 2025 Myanmar earthquake reveals a sharp, short-lived spike in public attention, followed by a rapid decay and long-tail persistence. As shown in Figure 4, the volume of Weibo posts surged immediately after the event occurred on March 28, peaking within the first 24 hours. More than 10,000 posts were published on March 29 alone, accounting for over 35% of the entire dataset. The hourly inset plot for the first week further illustrates that the peak intensity occurred within the first 6-12 hours post-event, aligning with the typical dynamics of sudden-onset disasters in social media discourse.

Fig. 4 Temporal distribution of Weibo posts related to the 2025 Myanmar Earthquake.

Following the initial surge, attention decreased rapidly, yet a smaller second wave appeared around March 30 - 31, likely driven by emerging reports of casualties and international rescue responses. After April 2, the volume of posts stabilized at a low level, indicating the event's

transition from acute response to long-term memory. This temporal profile mirrors patterns observed in previous studies of transboundary disasters, where digital attention is front-loaded and highly sensitive to real-time developments.

To complement this macro-level trend analysis, we further examined the structural and behavioral characteristics of the posts and their authors, as visualized in Figure 5. The upper row shows the distribution of four continuous variables: number of followers, number of followers, total posts by user (weibos), and post character count. All exhibit heavy-tailed distributions, suggesting that a small group of highly active or influential users contribute disproportionately to content production. The cumulative distribution functions (red dashed lines) indicate that over 80% of posts come from users with fewer than 500 followers, underscoring the grassroots nature of much of the discourse.

Fig. 5 Descriptive statistics of post features, user types, geographical distribution, and interaction correlations.

The bottom-left bar charts show the distribution of user verification types and geographical origins. Most posts originated from regular users and celebrities, with relatively smaller contributions from government and media accounts. Geographically, the top contributing

provinces were Guangdong, Beijing, Sichuan, and Yunnan, regions with high population density or geographical proximity to Myanmar. This suggests both demographic and spatial factors influenced participation in disaster discourse. And the correlation heatmap in the bottom-right corner displays the relationships among reposts, comments, and likes. While reposts and comments show a moderate correlation (r=0.40), their associations with likes are notably weaker, indicating that engagement forms are partially decoupled and may reflect different motivations, such as amplification versus emotional expression. Together, these descriptive results provide a comprehensive overview of the who, when, and where of public engagement with the Myanmar earthquake on Chinese social media.

#### 4.2 Emotion Evolution and Flow Patterns

Public emotional response to the 2025 Myanmar earthquake exhibited marked heterogeneity in both intensity and trajectory across seven classified emotions. As shown in Figure 6, the dominant emotional category throughout the observation window was Surprise, which remained consistently high from the onset of the event to the end of April. This is consistent with prior findings that transboundary disasters, particularly those perceived as sudden and foreign, tend to evoke shock-oriented reactions in early discourse phases.

Fig. 6 Temporal evolution of all seven classified emotions (Surprise, Sadness, Fear, Happiness, Good, Anger, Disgust) over the 30-day observation period.

Sadness ranked second in frequency, maintaining a stable presence with minor peaks around April 7 and April 21, potentially reflecting reports of increased casualties or stories of humanitarian loss. Fear, though initially declining, experienced a moderate resurgence mid-month,

suggesting a secondary wave of anxiety likely tied to aftershock speculation or cross-border concerns.

Notably, Happiness and Good, two positively valenced emotions, were also present, albeit at lower levels. Their increases in the second half of the period may correspond to hopeful developments such as successful rescues or international aid interventions. In contrast, Anger and Disgust remained marginal throughout the entire period, surfacing only intermittently. This distribution pattern reinforces the interpretation that Chinese public sentiment toward the Myanmar earthquake was largely characterized by empathetic rather than confrontational emotional framing.

To better capture the transitions among emotional states, we constructed a Sankey diagram based on daily emotion dominance during the first 10 days following the earthquake (see Figure 7). The restriction to this 10-day window reflects the front-loaded structure of online attention, during which emotional volatility was most pronounced.

Fig. 7 Emotion transition flows over the first 10 days following the Myanmar earthquake, represented as a Sankey diagram.

The Sankey flows illustrate how emotions shifted across days and into one another. Early transitions from Surprise to Sadness, and Fear to Sadness, suggest a cognitive-emotional reorientation from immediate shock toward grief and concern. A moderate number of flows also move from Sadness to Good or Happiness, particularly between April 2 and April 4, likely signaling public acknowledgment of effective response efforts. Emotional recycling is also visible, for example, loops from Sadness back to Fear, or from Surprise to Surprise, indicate that some

emotions are self-sustaining within public discourse.

Importantly, even less frequent emotions such as Disgust and Anger are captured as peripheral threads in the diagram, highlighting their low but non-zero rhetorical roles. Overall, the pattern of affective flow reveals a temporally structured and emotionally layered public response, shaped not only by factual developments but also by semantic framing, moral attribution, and transnational empathy.

### 4.3 Semantic Structure of Entity Co-occurrence

To explore the semantic scaffolding of public discourse, we constructed a large-scale entity co-occurrence network based on named entities extracted from the Weibo corpus. As shown in Figure 8, each node represents a unique named entity, and edges connect entities that co-occurred in at least one post. The network comprises seven categories of entities: Event/Outcome, Geography, Organization, Person, Time, Quantity/Value, and Other/Object, each visualized in a distinct color.

Fig. 8 Overall entity co-occurrence network colored by entity category (Event/Outcome, Geography, Organization, Person, Time, Quantity/Value, Other/Object).

The resulting network displays clear high-density clustering, particularly in the center, suggesting a concentrated set of entities that serve as semantic anchors throughout the discussion.

Notably, Event/Outcome and Geography entities form the densest cores, indicating that both the disaster itself and its spatial referents are key organizing axes of public attention. Peripheral clusters, often composed of Organization and Time entities, are linked through these central domains, implying a radial semantic architecture where institutional actors and temporal markers gain salience through their proximity to core events and locations.

To further examine category-specific patterns, we visualized four representative subnetworks in Figure 9, corresponding to Event/Outcome, Geography, Organization, and Time.

Fig. 9 Subnetworks for four representative entity categories.

The Event/Outcome network is centered on high-frequency nodes such as "earthquake," "collapse," and "casualties." These nodes exhibit high weighted degree and serve as bridges to

402

404

407

413414

discourse. In the Geography subnetwork, prominent nodes include both domestic provinces (e.g., Yunnan, Sichuan) and foreign regions (e.g., Sagaing Region, Mandalay City). The presence of cross-border geographic references reflects the transnational nature of the disaster and the spatial extension of empathetic concern. The Organization subnetwork is densely connected around rescue-related institutions and agencies, including national emergency bureaus, humanitarian organizations, and medical teams. These entities are tightly interlinked and often co-occur with Event/Outcome terms, underscoring a discourse of operational response and cross-agency mobilization. In the Time subnetwork, temporal anchors such as "March 28," "48 hours," and "April 1st" organize discourse sequences and associate specific emotional tones with event chronology. The prevalence of time-stamped expressions reveals the public's effort to construct temporal coherence in understanding the event's progression. Overall, these entity networks demonstrate how meaning is structurally embedded in co-reference patterns. They form the semantic substrate upon which emotional signals are projected and propagated, setting the stage for the emotion-entity coupling analysis in the next section. 4.4 Emotion Projection onto Entity Networks To move beyond conventional sentiment trendlines and capture the relational dynamics of emotion transmission, we constructed an emotion-entity coupled network by projecting classified emotional states onto the previously generated named entity co-occurrence structure. As shown in Figure 10, the resulting graph retains the topological backbone of entity interactions while encoding the dominant emotion associated with each entity pair in the form of color-coded and weighted edges. Edge color indicates the prevailing emotional tone, such as Surprise, Sadness, or Fear, while edge thickness corresponds to the frequency of emotional co-occurrence, thereby highlighting not just whether but how strongly specific emotions flow along semantic connections.

both institutional and geographic references, indicating their role as the narrative backbone of risk

Fig. 10 Emotion-entity coupled network, where edge color indicates dominant emotion and edge width reflects emotional co-occurrence frequency between named entities.

Several key features emerge from this enriched semantic—affective topology. First, Fear and Surprise dominate the overall network, often radiating from core disaster-related nodes such as "Myanmar earthquake," "epicenter," "rescue," and "Yunnan." These entities serve as emotionally resonant hubs, anchoring the interpretive and affective architecture of the network. The pervasiveness of Fear is particularly notable in connections involving geopolitical references (e.g., "Thailand," "Bangkok"), suggesting heightened concern for regional spillover effects or secondary threats. Surprise, in contrast, often binds structural terms (e.g., "occur," "strong earthquake") to actor-based entities like "reporter" or "Weibo Video," reflecting the narrative rhythm of breaking news and real-time witnessing.

Second, positive emotions, though less frequent, are not absent. Edges labeled with Good and Happiness cluster around terms like "Chinese rescue team," "CCTV news," and "aid," forming localized zones of hopeful discourse. These nodes not only reflect perceived effectiveness of

institutional response but also serve as affective counterweights to dominant negative emotion clusters, enabling moments of empathy, trust, and even pride to emerge amid the broader disaster narrative.

However, due to the high density of the full graph, interpretive clarity necessitates localized magnification. To this end, we extracted two semantically distinct subgraphs, displayed in Figure 11, to illustrate contrasting emotional patterns.

Fig. 11 Two zoomed-in subgraphs from the emotion-entity network highlighting (lower) an early-phase news-surprise cluster, and (upper) an institutional empathy cluster involving historical reference and humanitarian actors.

The first subgraph focuses on the cluster around "Myanmar earthquake," "Yunnan," and "Weibo Video." This region is characterized by a predominance of Surprise and Sadness, with frequent ties to reporting-related entities. The visual density of green and light blue edges reflects the rapid information flow and emotional ambivalence of the early response phase, where knowledge and uncertainty coexist.

The second subgraph captures a different dynamic: emotion pathways anchored in institutional and historical memory. Terms such as "Chinese rescue team," "Wenchuan Earthquake," and "CCTV news" are bound together through Happiness, Good, and Sadness, indicating not only

the intertextual linkage of past disasters but also the emotional anchoring of national response efforts. The strong edge between "Chinese rescue team" and "person" (colored in dark blue) exemplifies how organizational trust and human empathy are co-constructed within affective discourse.

These findings support a conceptual shift from viewing emotion as an aggregate metric to treating it as a distributed, structurally situated phenomenon. Emotions in disaster discourse are not merely reactions but are routed through, shaped by, and reinforced within semantic linkages. By tracing these paths, we capture how empathy is stabilized, how fear is amplified, and how institutional narratives modulate affective response in transboundary risk scenarios.

### 5. Discussion

The findings of this study offer a critical departure from conventional sentiment analysis paradigms in disaster communication by revealing how emotional meaning is constructed, transmitted, and stabilized through the underlying semantic structure of discourse. Rather than treating emotion as a static or disembodied category affixed to posts, we approached it as a structurally embedded phenomenon, dynamically routed through named entity interactions and co-reference patterns. This reorientation from aggregate sentiment statistics to relational emotion modeling allows for a more nuanced understanding of how public responses to transboundary disasters are affectively organized and semantically sustained.

At the center of this reconceptualization is the construction of a coupled emotion – entity network, which bridges two previously parallel lines of research: affective computing and disaster informatics. While prior studies have separately examined sentiment dynamics or entity recognition, they have rarely asked how these two dimensions interact (Yang et al., 2024; Xi et al., 2024). Our approach shows that emotions do not merely accompany messages, they are channeled through specific semantic infrastructures, with certain entities functioning as emotional hubs, others as amplifiers or buffers, and still others as neutral scaffolds. In particular, entities related to geographic proximity (e.g., "Yunnan," "Bangkok"), institutional actors (e.g., "Chinese rescue team," "CCTV News"), and historical referents (e.g., "Wenchuan Earthquake") emerged as key junctions through which both negative and positive emotions circulated. This structural positioning reveals the mechanisms by which empathic identification with a foreign disaster is

socially constructed in digital space.

What distinguishes this study is its attention to the differentiated affective topology of disaster discourse. The results demonstrate that emotions such as fear, surprise, and sadness do not distribute uniformly across entities, nor do they follow simple temporal decay or escalation curves. Instead, they cluster around particular semantic motifs and evolve along discrete narrative arcs. For example, fear dominates connections between event descriptors and international locations, while positive emotions such as good and happiness concentrate in networks involving institutional response and humanitarian relief. These affective asymmetries underscore the interpretive labor embedded in disaster sensemaking: the public does not respond to "the disaster" in the abstract, but to situated frames populated by identifiable agents, places, and outcomes. The emergence of empathic zones, localized clusters where emotional coherence is stabilized, illustrates how affect is both semantically patterned and socially negotiated.

Furthermore, our analysis reveals that emotion in crisis discourse is neither spontaneous nor arbitrary, but structured by recognizable mechanisms of resonance, proximity, and anchoring. The identification of dominant emotional pathways across the entity network makes visible how sentiments shift, intensify, or diffuse in relation to evolving public narratives (Zhong et al., 2025; Li, 2025). In this view, emotion is not merely a response but also a relational outcome, a product of how meaning circulates through inter-entity connections, and how the salience of particular nodes shapes collective emotional trajectories. This stands in contrast to models that treat emotion as a discrete, user-generated signal and opens a new methodological space for mechanism-oriented disaster research.

Our findings also contribute to the emerging literature on transboundary disaster empathy, particularly in the context of digital nationalism and regional geopolitics. The 2025 Myanmar earthquake did not occur on Chinese soil, yet it provoked intense emotional engagement across Chinese social media. This suggests that national boundaries do not limit affective publics, but rather serve as a terrain for constructing digital proximity and moral relevance. In this sense, emotional responses are not merely reflexive but are embedded within broader geopolitical, cultural, and historical matrices. Entities such as "Myanmar," "Chinese rescue team," and "Red Cross" do not simply name referents, they mobilize frames of responsibility, solidarity, and risk anticipation.

The interplay between semantic structure and emotional projection thus offers an analytical lens through which to assess not just what the public feels, but how feeling itself is organized and made possible through discourse. It also underscores the importance of considering the semantic architecture of social media narratives in evaluating public sentiment, particularly when dealing with low-visibility, high-impact disasters that cross national boundaries. Our emotion – entity coupling framework provides a flexible and generalizable method for capturing these dynamics, with potential applications in real-time disaster monitoring, humanitarian communication design, and digital diplomacy.

Finally, by foregrounding emotion as a networked and meaning-dependent phenomenon, this study contributes to a broader theoretical shift in disaster research: one that moves away from measuring affect as an input or outcome variable, and toward understanding it as an integral relational process. Emotions here are not noise to be filtered out of communication but signal-rich elements of how publics interpret, evaluate, and act upon crises.

## 6. Conclusion

This study has introduced a novel framework for analyzing emotional responses to transboundary disasters by coupling sentiment classification with named entity recognition on large-scale social media data. Using the 2025 Myanmar earthquake as a case, we demonstrated how public emotion on Chinese Weibo evolved not only temporally but also structurally, flowing through a dense web of named entities that functioned as semantic and affective anchors. Moving beyond traditional sentiment trendlines, our emotion – entity coupling approach allowed us to model emotional propagation as a networked phenomenon, revealing the co-dependence of meaning and feeling in digital disaster discourse.

Our findings underscore three core contributions. (1) We offer a methodological innovation

Our findings underscore three core contributions. (1) We offer a methodological innovation by aligning emotion analysis with entity-level semantic structure. This shift enables a more granular and mechanistic understanding of how emotion is situated within discourse, rather than abstracted from it. (2) We show that emotional responses are topologically organized: certain entities, especially those tied to location, institutional actors, and temporal markers, serve as key conduits for emotional diffusion. Rather than functioning in isolation, these nodes interact within affective circuits that reflect public meaning-making in the face of uncertainty and risk. (3) Our

546547

555

556557

565

humanitarian response, and spatial proximity, which are suggested that transnational affective publics emerge through the strategic convergence of language, history, and trust. The practical implications of our research are equally significant. Emotion - entity mapping can inform real-time crisis monitoring systems by identifying not only which sentiments dominate, but how they are structured and to whom they are attached. This has value for both governmental agencies and humanitarian organizations seeking to understand and respond to public sentiment during crises. Moreover, our findings provide insight into how empathy can be amplified, trust reinforced, or misinformation countered, through the targeted engagement of emotionally resonant entities within disaster narratives. While our study offers new pathways for analyzing emotion in digital disaster discourse, several limitations remain. The current framework does not yet account for user interaction dynamics such as retweets, mentions, or comment threads, which may influence how emotion circulates socially. In addition, while we employed a dual-channel annotation strategy combining LLM and human review, future research could further improve model precision through multimodal sentiment detection and deeper context modeling. **Declaration of Competing Interests** The authors declare that they have no known competing financial interests or personal relationships that could have appeared to influence the work reported in this paper. **Authorship contributions** CD: Conceptualization, Investigation, Methodology, Software, Visualization, Writing – original draft, Writing – review & editing. **KZ**: Conceptualization, Investigation, Methdology, Supervision. JL: Conceptualization, Methodology, Software, Visualization, Writing - review & editing, Project administration, Supervision.

study highlights the dynamics of digital empathy in cross-border contexts. The emotional salience

attributed to a foreign disaster is shaped by interlinked references to national memory,

International

Journal

of

| 567 | Funding                                                                                                             |
|-----|---------------------------------------------------------------------------------------------------------------------|
| 568 | This research was supported by the National Natural Science Foundation of China (Grant No.                          |
| 569 | 72504070; 72374056).                                                                                                |
| 570 |                                                                                                                     |
| 571 | Data availability                                                                                                   |
| 572 | Data will be made available on request.                                                                             |
| 573 |                                                                                                                     |
| 574 | References                                                                                                          |
| 575 |                                                                                                                     |
| 576 | Alexander, D. E.: Social media in disaster risk reduction and crisis management. Science and Engineering Ethics,    |
| 577 | 20(3), 717–733. https://doi.org/10.1007/s11948-013-9502-z, 2014.                                                    |
| 578 | Bai, H., and Yu, G.: A Weibo-based approach to disaster informatics: incidents monitor in post-disaster situation   |
| 579 | via Weibo text negative sentiment analysis. Natural Hazards, 83(2), 1177-1196.                                      |
| 580 | https://doi.org/10.1007/s11069-016-2370-5, 2016.                                                                    |
| 581 | Börner, S.: Emotions matter: EMPOWER-ing youth by integrating emotions of (chronic) disaster risk into              |
| 582 | strategies for disaster preparedness. International Journal of Disaster Risk Reduction, 89, 103636.                 |
| 583 | https://doi.org/10.1016/j.ijdrr.2023.103636, 2020.                                                                  |
| 584 | Castellanos, L. A, Edjossan-Sossou, A., and Komendantova, N.: Evolving Emotions: Tracing Social Media               |
| 585 | Narratives in the Wake of the Manchester Arena Bombing. International Journal of Disaster Risk Reduction,           |
| 586 | 129, 105722. https://doi.org/10.1016/j.ijdrr.2025.105722, 2025.                                                     |
| 587 | Chen, Y., Li, Y., Wang, Z., Quintero, A. J., Yang, C., and Ji, W.: Rapid perception of public opinion in emergency  |
| 588 | events through social media. Natural Hazards Review, 23(2), 04021066.                                               |
| 589 | https://doi.org/10.1061/(ASCE)NH.1527-6996.0000547, 2022.                                                           |
| 590 | Chu, M., Song, W., Zhao, Z., Chen, T., and Chiang, Y. C. (2024). Emotional contagion on social media and the        |
| 591 | simulation of intervention strategies after a disaster event: A modeling study. Humanities and Social Sciences      |
| 592 | Communications, 11(1), 968. https://doi.org/10.1057/s41599-024-03397-4, 2024.                                       |
| 593 | Dai, J., Zhao, Y., and Li, Z. Sentiment-topic dynamic collaborative analysis-based public opinion mapping in        |
| 594 | aviation disaster management: A case study of the MU5735 air crash. International Journal of Disaster Risk          |
| 595 | Reduction, 102, 104268, 968. https://doi.org/10.1016/j.ijdrr.2024.104268, 2024.                                     |
| 596 | Eligüzel, N., Çetinkaya, C., and Dereli, T.: Application of named entity recognition on tweets during earthquake    |
| 597 | disaster: a deep learning-based approach. Soft Computing, 26(1), 395-421.                                           |
| 598 | https://doi.org/10.1007/s00500-021-06370-4, 2022.                                                                   |
| 599 | Erokhin, D., and Komendantova, N.: Social media data for disaster risk management and research. International       |
| 600 | Journal of Disaster Risk Reduction, 114, 104980. https://doi.org/10.1016/j.ijdrr.2024.104980, 2024.                 |
| 601 | Gay, M.: Speaking to the self or others? Exploring existential, relational, and narrative identity in suicide notes |
| 602 | using thematic and psychodynamic analysis. Death Studies, 29, 1-15, 968.                                            |
| 603 | https://doi.org/10.1080/07481187.2025.2537975, 2025.                                                                |
| 604 | George, A. S., and Baskar, T.: Leveraging big data and sentiment analysis for actionable insights: A review of data |
| 605 | mining approaches for social media. Partners Universal International Innovation Journal, 2(4), 39-59.               |
| 606 | https://doi.org/10.5281/zenodo.13623777, 2024.                                                                      |
| 607 | George, L., and Sumathy, P.: An integrated clustering and BERT framework for improved topic modeling.               |

Technology,

15(4),

2187-2195.

- https://doi.org/10.1007/s41870-023-01268-w, 2023.
- Han, X., Wang, J.: Modelling and analyzing the semantic evolution of social media user behaviors during disaster
- events: A case study of COVID-19. ISPRS International Journal of Geo-Information, 11(7), 373.
- https://doi.org/10.3390/ijgi11070373, 2022.
- Hu, Y., Mai, G., Cundy, C., Choi, K., Lao, N., Liu, W., Lakhanpal, G., Zhou, R., and Joseph, K.:
- Geo-knowledge-guided GPT models improve the extraction of location descriptions from disaster-related
- social media messages. International Journal of Geographical Information Science, 37(11), 2289-2318.
- https://doi.org/10.1080/13658816.2023.2266495, 2023.
- Jin, Y., Liu, B. F., and Austin, L. L.: Examining the role of social media in effective crisis management: The effects
- of crisis origin, information form, and source on publics' crisis responses. Communication Research, 41(1),
- 74-94. https://doi.org/10.1177/0093650211423918, 2011.
- Kaufhold, M. A., and Reuter, C.: The self-organization of digital volunteers across social media: The case of the
- 2013 European floods in Germany. Journal of Homeland Security and Emergency Management, 13(1),
- 137-166. https://doi.org/10.1515/jhsem-2015-0063, 2016.
- Khurana, D., Koli, A., Khatter, K., and Singh, S.: Natural language processing: state of the art, current trends and
- challenges. Multimedia Tools and Applications, 82(3), 3713-3744.
- https://doi.org/10.1007/s11042-022-13428-4, 2023.
- Lee, M. J., Lee, T. R., Lee, S. J., Jang, J. S., and Kim, E. J.: Machine learning-based data mining method for
- sentiment analysis of the Sewol Ferry disaster's effect on social stress. Frontiers in Psychiatry, 11, 505673.
- https://doi.org/10.3389/fpsyt.2020.505673, 2020.
- Li, N.: Analyzing the complexity of public opinion evolution on weibo: A super network model. Journal of the
- Knowledge Economy, 16(1), 3404-3439. https://doi.org/10.1007/s13132-024-02059-9, 2025.
- Li, S., Liu, Z., and Li, Y.: Temporal and spatial evolution of online public sentiment on emergencies. Information
- processing & management, 57(2), 102177. https://doi.org/10.1016/j.ipm.2019.102177, 2020.
- Li, Z., and Zhang, X.: Research on named entity recognition methods for urban underground space disasters based
- on text information extraction. The International Archives of the Photogrammetry, Remote Sensing and
- Spatial Information Sciences, 48, 547-552.
- https://doi.org/10.5194/isprs-archives-XLVIII-1-W2-2023-547-2023, 2023.
- Liu, J., Liu, W., Yan, C., and Liu, X.: Study on the temporal and spatial evolution characteristics of chinese public's
- cognition and attitude to "double reduction" Policy based on big data. Big Data Research, 34, 100411.
- https://doi.org/10.1016/j.bdr.2023.100411, 2023.
- Lüders, A., Dinkelberg, A., and Quayle, M.: Becoming "us" in digital spaces: How online users creatively and
- strategically exploit social media affordances to build up social identity. Acta Psychologica, 228, 103643.
- https://doi.org/10.1016/j.actpsy.2022.103643, 2022.
- Lwowski, B., Rad, P., and Choo, K. K. R.: Geospatial event detection by grouping emotion contagion in social
- media. IEEE Transactions on Big Data, 6(1), 159-170. https://doi.org/10.1109/TBDATA.2018.2876405, 2018.
- Ma, B., and Zheng, R.: Exploring food safety emergency incidents on Sina Weibo: using text mining and sentiment
   evolution. Journal of Food Protection, 88(1), 100418. https://doi.org/10.1016/j.jfp.2024.100418, 2025.
- 647 Ma, X., Liu, W., Zhou, X., Qin, C., Chen, Y., Xiang, Y., Zhang, X., and Zhao, M.: Evolution of online public
- opinion during meteorological disasters. Environmental Hazards, 19(4), 375-397
- https://doi.org/10.1080/17477891.2019.1685932, 2020.
- Ma, Z., Li, L., Mao, Y., Wang, Y., Patsy, O. G., Bensi, M. T., Hemphill, L., and Baecher, G. B.: Surveying the use
- of social media data and natural language processing techniques to investigate natural disasters. Natural
- Hazards Review, 25(4), 03124003. https://doi.org/10.1061/NHREFO.NHENG-2047, 2024.

- Mirbabaie, M., Bunker, D., Stieglitz, S., Marx, J., and Ehnis, C.: Social media in times of crisis: Learning from 654
- Hurricane Harvey for the coronavirus disease 2019 pandemic response. Journal of Information Technology,
- 35(3), 195-213. https://doi.org/10.1177/0268396220929258, 2020.
- Misuraca, M., Scepi, G., and Spano, M.: A network-based concept extraction for managing customer requests in a 656
- social media care context. International Journal of Information Management, 51, 101956. 658 https://doi.org/10.1016/j.ijinfomgt.2019.05.012, 2020.
- Murdock, I., Carley, K. M., and Yağan, O.: An agent-based model of cross-platform information diffusion and
- moderation. Social Network Analysis and Mining, 14(1), 145. https://doi.org/10.1007/s13278-024-01305-x, 661
- Qiu, Q., Huang, Z., Xu, D., Ma, K., Tao, L., Wang, R., Chen, J., Xie, Z., and Pan, Y.: Integrating NLP and ontology
- matching into a unified system for automated information extraction from geological hazard reports. Journal
- of Earth Science, 34(5), 1433-1446. https://doi.org/10.1007/s12583-022-1716-z, 2023.
- Reuter, C., and Kaufhold, M. A.: Fifteen years of social media in emergencies: a retrospective review and future
- directions for crisis informatics. Journal of Contingencies and Crisis Management, 26(1), 41-57.
- https://doi.org/10.1111/1468-5973.12196, 2018.
- Samatan, N., Fatoni, A., and Murtiasih, S.: Disaster communication patterns and behaviors on social media: a
- study social network# BANJIR2020 on Twitter. Humanities and Social Sciences Reviews, 8(4), 27-36.
- https://doi.org/10.18510/hssr.2020.844, 2020.
- Shapouri, S., Soleymani, S., and Rezayi, S.: Flood of techniques and drought of theories: emotion mining in
- disasters. Journal of Computational Social Science, 8(1), 5. https://doi.org/10.1007/s42001-024-00330-2,
- 2025.
- Singh, J., and Singh, A. K.: Twitter emergency response system during flood-related disaster. International Journal
- of Emergency Management, 17(2), 177-193. https://doi.org/10.1504/IJEM.2021.122931, 2023.
- Stella, M.: Cognitive network science for understanding online social cognitions: A brief review. Topics in 677 Cognitive Science, 14(1), 143-162. https://doi.org/10.1111/tops.12551, 2022.
- Stieglitz, S., Mirbabaie, M., Ross, B., and Neuberger, C.: Social media analytics-Challenges in topic discovery,
- data collection, and data preparation. International Journal of Information Management, 39, 156-168.
- https://doi.org/10.1016/j.ijinfomgt.2017.12.002, 2018.
- Storey, V. C., and O'Leary, D. E.:Text analysis of evolving emotions and sentiments in COVID-19 Twitter
- communication. Cognitive Computation, 16(4), 1834-1857. https://doi.org/10.1007/s12559-022-10025-3, 682
- 2024.
- Sun, J., Liu, Y., Cui, J., and He, H.: Deep learning-based methods for natural hazard named entity recognition.
- Scientific Reports, 12(1), 4598. https://doi.org/10.1038/s41598-022-08667-2, 2022.
- Tim, Y., Pan, S. L., Ractham, P., and Kaewkitipong, L.: Digitally enabled disaster response: the emergence of 686
- social media as boundary objects in a flooding disaster. Information Systems Journal, 27(2), 197-232.
- https://doi.org/10.1111/isj.12114, 2017.
- Venkatesan, S., Valecha, R., Yaraghi, N., Oh, O., and Rao, H. R.: Influence in Social Media: An Investigation of
- Spanning 2011 Egyptian Revolution. Ouarterly. 45(4).
- https://doi.org/10.25300/MISQ/2021/15297, 2021.
- Wang, W., Zhu, X., Lu, P., Zhao, Y., Chen, Y., and Zhang, S.: Spatio-temporal evolution of public opinion on urban
- flooding: Case study of the 7.20 Henan extreme flood event. International Journal of Disaster Risk Reduction,
- 100, 104175. https://doi.org/10.1016/j.ijdrr.2023.104175, 2024.
- Wilkho, R. S., and Gharaibeh, N. G.: FF-NER: A named entity recognition model for harvesting web-based
- information about flash floods and related infrastructure impacts. International Journal of Disaster Risk

# https://doi.org/10.5194/egusphere-2025-4507 Preprint. Discussion started: 14 November 2025 © Author(s) 2025. CC BY 4.0 License.

- Reduction, 105604. https://doi.org/10.1016/j.ijdrr.2025.105604, 2025.
- Xi, D., Zhou, J., Xu, W., and Tang, L.: Discrete emotion synchronicity and video engagement on social media: A
   moment-to-moment analysis. International Journal of Electronic Commerce, 28(1), 108-144.
   https://doi.org/10.1080/10864415.2023.2295072, 2024.
- Xu, Z., Lachlan, K., Ellis, L., and Rainear, A. M.: Understanding public opinion in different disaster stages: A case
   study of Hurricane Irma. Internet Research, 30(2), 695-709. https://doi.org/10.1108/INTR-12-2018-0517,
   2020.
- Yang, J., Zhang, T., Tsai, C. Y., Lu, Y., and Yao, L.: Evolution and emerging trends of named entity recognition:
  Bibliometric analysis from 2000 to 2023. Heliyon, 10(9), e30053.
  https://doi.org/10.1016/j.heliyon.2024.e30053, 2024.
- Yin, F., Tang, X., Liang, T., Kuang, Q., Wang, J., Ma, R., Miao, F., and Wu, J.: Coupled dynamics of information propagation and emotion influence: Emerging emotion clusters for public health emergency messages on the
   Chinese Sina Microblog. Physica A: Statistical Mechanics and its Applications, 639, 129630.
   https://doi.org/10.1016/j.physa.2024.129630, 2024.
- Zhang, K., Dong, C., Guo, Y., Yu, G., and Mi, J.: Intelligent Digital Twin for Predicting Technology Discourse
   Patterns: Agent-Based Modeling of User Interactions and Sentiment Dynamics in DeepSeek Discourse Case.
   Systems, 13(6), 451. https://doi.org/10.3390/systems13060451, 2025a.
- Zhang, K., Dong, C., Guo, Y., Zhou, W., Yu, G., and Mi, J.: Lagged Stance Interactions and Counter-Spiral of
   Silence: A Data-Driven Analysis and Agent-Based Modeling of Technical Public Opinion Events. Systems,
   13(6), 417. https://doi.org/10.3390/systems13060417, 2025b.
- Zhang, P., Zhang, H., and Kong, F.: Research on online public opinion in the investigation of the "7-20"
   extraordinary rainstorm and flooding disaster in Zhengzhou, China. International Journal of Disaster Risk
   Reduction, 105, 104422. https://doi.org/10.1016/j.ijdrr.2024.104422, 2024.
- Zheng, P., Adams, P. C., and Wang, J.: Shifting moods on Sina Weibo: The first 12 weeks of COVID-19 in Wuhan.
   New Media & Society, 26(1), 346-367. https://doi.org/10.1177/14614448211058850, 2021.
- Zhong, J., Mo, Y., Zhang, J., Liu, P., Luo, X., Liu, L., Ding, R., Huang, J., and Zheng, Y.: Beyond anger:
   uncovering complex emotional patterns between cyberbullying roles through affective computing and
   epistemic network analysis. Humanities and Social Sciences Communications, 12(1), 1281.
   https://doi.org/10.1080/10864415.2023.2295072, 2025.

726