# Peer review of "Tracing Emotional Evolution along Named Entity Topic"

_EGUsphere, 2025_

## Referee Comment (RC2)

**Tracing Emotional Evolution along Named Entity Topic 2 Chains: A Mechanistic Study of Chinese Social Media in the 3 2025 Myanmar Earthquake**

Reviewer Notes 20.12.2025

**Abstract**

In the abstract, enhance clarity and conciseness, minimize metaphorical expressions ("flows," "conduits," "zones of empathic resonance"), and provide greater methodological detail.

**Introduction**

The introduction is excessively long and contains repetitive critiques of traditional sentiment analysis. Multiple paragraphs reiterate points such as the overly aggregate nature of sentiment and the lack of semantic grounding, without providing additional nuance. The writing sometimes emphasizes theoretical elegance at the expense of analytical clarity, which may challenge reader engagement. Furthermore, the term "mechanism" is not clearly defined early in the text. While the introduction is generally persuasive, it would benefit from greater conciseness, a more precise problem definition, and improved conceptual focus.

**Literature review**

The literature review demonstrates competence and currency; however, it requires a more focused scope, clearer conceptual boundaries, and more selective critical analysis.

**Methodology**

The overall workflow, encompassing data collection, annotation, model fine-tuning, and network construction, is logical and well structured. However, the absence of quantitative validation for both emotion classification and named entity recognition (NER), such as F1 score, accuracy, or inter-annotator agreement, is a significant limitation. Assigning a single "dominant emotion" to each post oversimplifies the emotional complexity present in disaster discourse. The inference of "emotion transitions" based solely on entity co-occurrence lacks formal justification and may conflate association with propagation. Furthermore, key modeling decisions, including edge weighting, threshold selection, and network pruning, are not sufficiently justified. In summary, while the methodology is ambitious and well-designed at a conceptual level, it lacks adequate validation, transparency, and robustness checks to fully substantiate the paper's mechanistic claims.

**Results**

The results are presented in a clear sequence, moving from descriptive statistics to emotional dynamics and ultimately to semantic–affective networks. Nevertheless, several significant limitations are evident. The findings are predominantly descriptive and visual, with minimal quantitative testing or statistical validation. Assertions regarding "emotion flow," "amplification," and "buffering" are based on co-occurrence

patterns rather than being formally substantiated. The extensive use of complex network visualizations increases the risk of interpretive overreach and reader subjectivity. Furthermore, the lack of baselines or null models hinders the assessment of whether the observed patterns are distinctive or could occur by chance. Although positive emotion clusters are emphasized, their relative magnitude and robustness are not systematically quantified. In summary, while the results are comprehensive, coherent, and visually engaging, they remain largely exploratory. The strength of interpretive claims is not consistently supported by analytical rigor.

**Discussion**

The discussion appears to overinterpret descriptive findings, particularly when inferring mechanisms such as amplification and buffering without formal causal evidence. While claims regarding nationalism, trust construction, and geopolitical effect are plausible, they remain untested empirically. Alternative explanations, including media agenda-setting, platform effects, censorship, and posting norms, receive insufficient consideration. The discussion reiterates conceptual contributions at length, resulting in redundancy. Although the discussion is thoughtful and theoretically ambitious, it extends beyond what the results can robustly support. Furthermore, only two references are cited in the discussion, which limits its scientific support.

**Conclusion**

The conclusion section reiterates claims from the Discussion but does not synthesize insights at a broader conceptual level. The use of mechanistic and causal language is overstated, given the analysis's primarily descriptive nature. Although limitations are acknowledged, their implications for interpretation are not thoroughly examined. Suggestions for future research are concise but lack specificity. Overall, while the conclusion is coherent and well written, it adds little beyond summarization.

**Reviewer Suggestions**

The authors' efforts in preparing this manuscript are appreciated.

The manuscript addresses a significant and timely topic in disaster management. However, several substantive concerns remain.

The study relies heavily on emotion classification but does not report essential validation metrics (e.g., accuracy, Scores, inter-annotator agreement). Without quantitative evidence of model performance, it is difficult to assess the reliability of the core analytical outputs on which the conclusions depend.

The manuscript frequently describes the analysis as "mechanistic" and interprets emotion as "flowing" or "propagating" through entity networks. However, the empirical foundation for these claims is primarily based on co-occurrence patterns rather than on formally specified mechanisms, causal inference, or diffusion modeling. This discrepancy results in a gap between the strength of the claims and the evidentiary support provided.

Although the results are rich and visually compelling, many interpretations, particularly those concerning emotional amplification, buffering, trust construction, and digital nationalism, extend beyond what can be robustly inferred from the analyses presented.

Key concepts such as emotion propagation, semantic routing, and affective pathways are used extensively, yet remain insufficiently operationalized. Consequently, the manuscript at times blurs the distinction between analytical metaphor and empirical demonstration.

Collectively, these issues necessitate substantial reconceptualization, additional validation, and methodological strengthening that exceed the scope of a standard revision process.

Thank you.

Regards.